# Mechanical Circulatory Support Strategies in Takotsubo Syndrome with Cardiogenic Shock: A Systematic Review

**DOI:** 10.3390/jcm13020473

**Published:** 2024-01-15

**Authors:** Johanna K. R. von Mackensen, Vanessa I. T. Zwaans, Ahmed El Shazly, Karel M. Van Praet, Roland Heck, Christoph T. Starck, Felix Schoenrath, Evgenij V. Potapov, Joerg Kempfert, Stephan Jacobs, Volkmar Falk, Leonhard Wert

**Affiliations:** 1Department of Cardiothoracic and Vascular Surgery, Deutsches Herzzentrum der Charité—Medical Heart Center of Charité and German Heart Institute, 13353 Berlin, Germany; zwaans@stud.uni-heidelberg.de (V.I.T.Z.); ahmed.el-shazly@dhzc-charite.de (A.E.S.); roland.heck@dhzc-charite.de (R.H.); christoph.starck@dhzc-charite.de (C.T.S.); felix.schoenrath@dhzc-charite.de (F.S.); evgenij.potapov@dhzc-charite.de (E.V.P.); joerg.kempfert@dhzc-charite.de (J.K.); stephan.jacobs@dhzc-charite.de (S.J.); volkmar.falk@dhzc-charite.de (V.F.); leonhard.wert@dhzc-charite.de (L.W.); 2Department of Cardiothoracic Surgery, ASZ Hospital Aalst, 9300 Aalst, Belgium; 3Cardiac Surgery Department, Hartcentrum OLV Aalst, 9300 Aalst, Belgium; 4DZHK (German Center for Cardiovascular Research), Partner Site, 10785 Berlin, Germany; 5Department of Cardiothoracic Surgery, Charité, Universitätsmedizin Berlin, Corporate Member of Freie Universität Berlin, Humboldt-Universität zu Berlin, and Berlin Institute of Health, 10117 Berlin, Germany; 6Department of Health Sciences and Technology, ETH Zürich, 8093 Zurich, Switzerland

**Keywords:** Takotsubo, stress cardiomyopathy, cardiogenic shock, mechanical circulatory support, ECLS, IABP, Impella

## Abstract

Background: Takotsubo syndrome is, by definition, a reversible form of acute heart failure. If cardiac output is severely reduced, Takotsubo syndrome can cause cardiogenic shock, and mechanical circulatory support can serve as a bridge to recovery. To date, there are no recommendations on when to use mechanical circulatory support and on which device is particularly effective in this context. Our aim was to determine the best treatment strategy. Methods: A systematic literature research and analysis of individual patient data was performed in MEDLINE/PubMed according to PRISMA guidelines. Our research considered original works published until 31 July 2023. Results: A total of 93 publications that met the inclusion criteria were identified, providing individual data from 124 patients. Of these, 62 (50%) were treated with veno-arterial extracorporeal life support (va-ECLS), and 44 (35.5%) received a microaxial left ventricular assist device (Impella). Eighteen patients received an Impella CP and twenty-one an Impella 2.5. An intra-aortic balloon pump (IABP) without other devices was used in only 13 patients (10.5%), while other devices (BiVAD or Tandem Heart) were used in 5 patients (4%). The median initial left ventricular ejection fraction was 20%, with no difference between the four device groups except for the IABP group, which was less affected by cardiac output failure (*p* = 0.015). The overall survival was 86.3%. Compared to the other groups, the time to cardiac recovery was shorter with Impella (*p* < 0.001). Conclusions: Though the Impella treatment is new, our analysis may show a significant benefit of Impella compared to other MCS strategies for cardiogenic shock in Takotsubo syndrome.

## 1. Introduction

Takotsubo syndrome (TTS) is an acute and reversible manifestation of heart failure, commonly found in patients presenting with symptoms leading to acute coronary syndrome (ACS), especially women [1,2]. The 2019 coronavirus pandemic, as a global event with major psychosocial impacts, resulted in a dramatic increase in the number of TTS cases and further demonstrated the importance of TTS as a differential diagnosis of ACS. During the COVID-19 pandemic, the incidence of patients presenting with ACS with subsequently confirmed TTS increased to 7.8%, from 3% previously [3,4]. However, TTS remains underdiagnosed, largely due to the interplay of acute coronary syndrome.

The incidence of cardiogenic shock (CS) in patients with TTS is estimated as ranging between 6% and 20% [5,6]. Patients with TTS in CS have a high in-hospital mortality of 15% [6]. Both the Mayo Clinic Criteria [7] and the InterTAK Diagnostic Criteria [8] are used as instruments for diagnosing TTS. They include echocardiographic evidence of hypo-, a- or dyskinesia extending beyond a single coronary vascular territory, as well as exclusion by coronary angiography of coronary atherosclerosis as a causal agent of regional wall motion abnormalities. Physical or emotional triggers should additionally be detectable. The main difference between the two criteria is that, for the Mayo Clinic Criteria, the absence of pheochromocytoma must be confirmed, whereas the InterTAK Diagnostic Criteria postulate that certain neurological disorders (e.g., subarachnoid haemorrhage, stroke/transient ischaemic attack or seizures) as well as pheochromocytoma may serve as triggers for TTS [7,8]. Even though research has paid increasing attention to TTS in recent years, many aspects remain unclear. Besides its aetiology, which is not completely understood, the management of patients with TTS, especially when a severe course of stress cardiomyopathy leads to cardiogenic shock, is unclear and there is no evidence-based therapy concept available. As mechanical circulatory support (MCS) devices are increasingly used in patients with CS and are recommended for SCAI shock stages C, D and E [9], they are also used in patients with TTS with good results [10,11,12].

A so-called broken heart can be repaired mechanically. However, there are no recommendations for when MCS should be used and which device should be preferred. The literature increasingly recommends considering MCS as a bridge-to-recovery strategy in TTS-CS [13,14], which by definition is reversible, in order to minimise the use of inotropes, as these may further worsen the clinical picture, which is potentially triggered by catecholamines [15].

Data on the use of MCS in TTS-CS are rare and only retrospective. The use of IABP was evaluated in a European registry of 2248 TTS patients, of whom 43 were patients with CS treated with IABP. The data show no significant difference in 30-day mortality, length of hospitalisation, need for invasive ventilation and 42-month follow-up [16]. A review and meta-analysis of MCS for TTS-CS by Mariani et al., including data until 2019, also confirms that the use of IABP is decreasing, while the use of va-ECLS and Impella in TTS patients with CS is increasing [17]. Since the microaxial LVAD Impella is a relatively new device, its frequent use is reflected only in the most recent literature; therefore, the review from Mariani et al. from 2020 included only 10 patients supported by Impella. The Impella is a non-durable microaxial left ventricular assist device (LVAD). Different surgical Impella procedures exist: the implantation of an Impella CP in combination with veno-arterial ECLS via the femoral artery is called ECMELLA 1.0 [18], while the combination of a single axillary artery approach for an Impella CP with veno-arterial ECLS is referred to as ECMELLA 2.0 [19]. The various procedures are associated with different complications (e.g., wound infection rates) and possibilities (e.g., early patient mobilisation) [20]. In recent years, it has been increasingly suggested that microaxial LVADs may be of particular value in the treatment of CS caused by TTS, as they enable venting of the left ventricle [21,22,23]. There is thus a growing interest to evaluate patients with TTS-CS who have been treated with Impella [24,25,26,27].

A retrospective multicentre analysis conducted in 2022 evaluated 16 patients with TTS-CS treated with Impella; the data indicated favourable outcomes, with good survival and full recovery of LV function [24].

In general, the literature contains reports on the use of intra-aortic balloon pumps (IABP), TandemHeart, extracorporeal membrane oxygenation (ECLS) and microaxial pumps (i.e., Impella) for the treatment of TTS-CS. Since interest in this clinical cardiac entity is increasing, there is also a rapidly growing number of published case reports and series relating to the use of MCS devices in TTS-CS. To recognise the trends and outcomes of this currently rapidly growing population of patients with TTS-CS treated with MCS and to compare the different support systems that can be applied, we performed a systematic review and analysis of the existing literature.

## 2. Materials and Methods

This review was conducted and reported in accordance with the Preferred Reporting Items for Systematic Reviews and Meta-Analyses (PRISMA) guidelines [28]. Prior to the literature review, the concept, inclusion criteria, research question and hypothesis were defined. The aim was to compare the different devices used so far as MSC strategies in the context of TTS-CS in the existing literature. The underlying hypothesis was that left ventricular venting and physiological antegrade support provided by the microaxial Impella device allows the overloaded ventricle to recover faster and that MCS is therefore needed for a shorter period of time.

### 2.1. Search Strategies

A systematic search in PubMed of articles published until 31 July 2023 was carried out using the following mesh terms: (‘takotsubo’ or ‘tako-tsubo’ or ‘stress cardiomyopathy’ or ‘apical ballooning’) AND (‘ECLS’ or ‘ECMO’ or ‘Impella’ or ‘IABP’ or ‘mechanical support’ or ‘mechanical circulatory support’ or ‘MCS’ or ‘assist device’). The titles of the initial results were screened and all publications matching the inclusion criteria of confirmed TTS and use of MCS were imported into the reference management software EndNote 20^®^ (Clarivate Analytics, London, UK). Duplicates were removed and the remaining publications were assessed for inclusion and screened for additional matching references. Full texts of all relevant articles were obtained and evaluated by the first author (Figure 1).

### 2.2. Inclusion Criteria

Published case reports and case series as well as retrospective, observational or randomised studies among patients diagnosed with TTS were considered. Inclusion required documentation of the following criteria:TTS diagnosis consistent with the InterTAK Diagnostic Criteria and designated as such by the authors;Use of temporary MCS: IABP, ECLS, Impella, TandemHeart;Individually reported patient data in terms of pre-intervention status, survival and time on support.

Studies not written in English or German, poorly described cases and review articles were excluded. Case series or studies not providing primary data or where the analysis was pooled without a description of individual patient data were excluded.

### 2.3. Data Extraction

Data were extracted by the first author and independently checked for accuracy by the last author. Data extracted included authors, title, year of publication, patient demographics (age, sex), MCS strategy, vital signs before circulatory support (systolic/diastolic blood pressure or MAP, heart rate), respiratory failure, use of inotropes before MCS, performed angiography (LVEDP), cardiac arrest before MCS, new ECG abnormalities (ST elevation, ST depression and T wave inversion), new arrhythmias, LVEF (before MCS, on MCS and in follow-up), NTproBNP/BNP (baseline and follow-up), baseline lactate, presence of left ventricular outflow tract obstruction (LVOTO) (gradient) or mitral regurgitation (MR) (systolic anterior motion), trigger, TTS type (inverted or normal), RV involvement, time to weaning of inotropes under MCS, time on MCS and follow-up (last documented date after implantation, LV recovery and survival). Patients treated with more than one device were assigned to a device group for statistical analysis (see Figure 2). The assignment was based on the device to which the authors attributed therapy success. Thus, if MCS therapy was escalated to another device under which therapy success was achieved, the patient was assigned to the device group to which it was escalated.

### 2.4. Statistics

Descriptive statistical analysis was performed by using the data analysis tools IBM^®^ SPSS 29 (IBM, Armonk, New York, NY, USA) and R, Version 4.02.(R Project for Statistical Computing, the R Foundation for Statistical Computing, Vienna, Austria).

The following data were collected: median with interquartile range (IQR) or mean ± SD for the total group and for the separate device subgroups. Categorical variables are presented as *N* (%) and were compared using Fisher’s exact test. Metric data were analysed for >2 independent variables using the Friedman test or the Kruskal–Wallis test depending on the distribution (normal or non-normal). A two-sided *p*-value lower than 0.05 was considered statistically significant. To enable assignment of significant results, a pairwise comparison was performed if more than two variables (e.g., four device groups) were used.

## 3. Results

### 3.1. Literature Research

The search in PubMed yielded a total of 1221 publications. A screening of the titles identified 1038 irrelevant or double publications. A total of 183 articles were screened on the basis of the abstract; of these, 141 were classified as relevant and full-text reading was carried out. A screening of the reference list yielded a further nine publications. Forty-nine articles did not meet the inclusion criteria or provide relevant information and were therefore excluded. Two articles depicted the same case, so one was excluded [30,31]. Seven articles could not be included because the full text was not available in English or German [32,33,34,35,36,37,38].

After full-text evaluation, 93 articles providing individual data on 124 cases were included (Appendix A). While the publications cover a period of almost 20 years, the majority were from the last five years (Figure 3).

The quality of case-related data in the publications varied widely. In particular, some articles were dedicated to specific aspects, e.g., echocardiographic control or therapy of the underlying condition, and therefore provided little general data. Overall, an almost complete data set could be collected for demographics, baseline LVEF and the duration of mechanical support. Especially information regarding follow-up and clinical status prior to the establishment of MCS, e.g., blood pressure, heart rate or catecholamines before MCS, varied greatly in terms of availability and quality. Unfortunately, important haemodynamic parameters such as right heart catheter findings or parameters of extended cardiac monitoring were not provided in the underlying literature.

### 3.2. Demographics and Device Groups

A data set of 124 patients was obtained, 92 of whom were female (74.2%). Five paediatric cases were included (age < 18 years). The median age was 52.2 years (Table 1).

After assigning the patients with two or more devices to a device group (Figure 2), the patient collective was divided into four device groups: ECLS, Impella, IABP and Other (Figure 4). The ‘Other’ group included two cases where a TandemHeart was combined with IABP and Impella, as well as one case where central ECLS was established and two cases treated with a BiVAD [39,40,41,42,43]. The Impella group was further subdivided according to the generation used: the Impella 2.5, which can increase cardiac output (CO) by about 2.5–3 L/min, was the most frequently used model, accounting for 47.7% of Impella cases. The Impella CP, which can relieve the left ventricle and support the circulation with a higher capacity of up to 4 L/min, was used in 40.9% of the cases. In 4% of the Impella cases, no exact model was named.

### 3.3. Pre-MCS Status of the Patients

In most cases, TTS was caused by physical triggers, but there were also cases in which a combination of physical and emotional triggers was present. Iatrogenic catecholamine administration was thought to be the cause of TTS-CS in six patients [40,44,45,46,47]. Nineteen patients developed TTS-CS in the context of pheochromocytoma [48,49,50,51,52,53,54,55,56,57,58,59,60], all of whom received ECLS as MCS, one in combination with IABP and one in combination with Impella. Due to pheochromocytoma, these patients exhibited a markedly higher blood pressure before MCS, with a mean systolic blood pressure of 140.9 mmHg. This is also reflected in the difference in the median systolic blood pressures between the different device groups, with the ECLS groups showing a median systolic blood pressure of 95 mmHg compared with 82 mmHg or lower in the other groups. Another difference in pre-implantation status between the groups was the frequency of cardiac catheterisation to rule out coronary obstruction. This is presumably attributable to the fact that both IABP as well as Impella 2.5 and CP are placed under angiographic support, which means that these patients inevitably visit the catheter laboratory; catheterisation is thus also mentioned in almost every case report. Another important difference is the presence of pulmonary oedema or respiratory failure. Patients who experienced a pulmonary complication or respiratory failure during CS more frequently received ECLS. There were no significant differences between the device groups with regard to cardiac arrest, ST elevations in the electrocardiogram (ECG), need for inotropes before implantation or elevated troponin levels.

Data between the groups regarding the clinical status of the patients before MCS therapy vary only for some parameters and are generally comparable (Table 2). However, for several parameters, data are only available from a fraction of the patients, making it difficult to draw conclusions as to the comparability of the groups. Regarding pre-implantation status, baseline LVEF is considered a key parameter for assessing the severity of cardiac compromise and the comparability of the severity of pump failure between the groups.

It should be emphasised that there was no significant difference in baseline LVEF across the various MCS groups. With a *p* value of 0.015, a significant result was found when comparing the median baseline LVEF values between the groups; however, a pairwise comparison showed that this significance existed only between IABP and the Impella as well as IABP and ECLS. The median values of baseline LVEF between the Impella and ECLS, however, did not differ significantly (*p*: 0.665).

### 3.4. Outcome and Follow-Up

In conclusion, the outcomes were good, with an overall survival rate of 86.3%. It should also be emphasised that in those cases where the patients died, they mostly suffered from very severe underlying diseases (two perioperative cases after liver transplant, one perioperative case after double-lung transplant, three cases with pheochromocytoma, two cases after polytrauma, one case with severe hypothermia and hypoglycaemia) [61,62,63,64,65,66]. In three cases, the authors did not attribute the patient’s death directly to the underlining condition. One suffered from advanced motor neuron disease and developed pneumonia and sepsis during MCS. One developed TTS after mitral valve reconstruction surgery. One case ‘merely’ experienced an emotional trigger but suffered from severe TTS; this patient died from the complications of MCS [67,68,69]. Significant differences between the groups, not caused by insufficient or inconsistent data (length of follow-up), were found only with regard to median support duration. Here, the Impella device clearly showed significantly shorter support times, which were also confirmed in pairwise comparison with every other device. Especially interesting is that the analysis of the Impella subgroups regarding the duration of MCS (Figure 5) showed that the duration of support decreased with the performance of the Impella generation. In other words: the greater the degree of LV venting, the faster the heart recovers and mechanical support can be terminated.

Unfortunately, it was hardly ever reported whether complications occurred during MCS therapy (Table 3). Furthermore, the recorded complications are not based on standardised definitions, but rather on what the authors of the primary literature report. In total, MCS complications were reported in 36 patients (29%). The occurrence of complications did not differ significantly between the groups (*p* = 0.927). Since several specific complications were mentioned in some cases and these were recorded, more individual complications are listed in some device groups in Table 4 than for the total number of patients suffering from complications in the device group. In a few publications, the persistence of cardiogenic shock under MCS was considered a complication, but because all patients in whom therapy was escalated to another device theoretically suffered from persistent CS under MCS, this category is not presented. Since the individual complication items are not based on common definitions, but merely on the fact that the authors of the case reports referred to them as such, and since complications under MCS were not mentioned in many articles, we deliberately refrained from performing a statistical evaluation of these parameters.

### 3.5. Comparison of the Device Groups

The collected data were analysed statistically with regard to the hypothesis that due to the pathophysiology of TTS-CS and the function of the different devices, the Impella is particularly suitable for circulatory support in TTS-CS. The data revealed that with comparable disease severity (especially baseline LVEF), patients in the Impella group required MCS for a significantly shorter period of time. However, there were no clear differences in the complication rates and overall survival (Figure 6).

## 4. Discussion

The patient population with Takotsubo syndrome is a very heterogeneous group, both in terms of concomitant diseases and the triggers of TTS. Some patients develop TTS within the scope of a severe non-cardiac condition and therefore have a critical status simply because of their underlying disease. Other patients, however, develop TTS within the scope of an emotional stressor and are physically healthy until they develop TTS. In addition, the diagnostic criteria that can be applied differ with respect to the inclusion of patients who develop TTS in the context of a neurological disease or pheochromocytoma. These patients are included when the InterTAK criteria are applied, thus generating a much more heterogeneous patient population, even if cardiac shock develops in the context of TTS, since the usual parameters such as low systolic pressure cannot be used to assess the severity of the clinical status in patients with pheochromocytoma. In addition to the very heterogeneous patient population, this evaluation is grounded on a very heterogeneous and retrospective data base. Despite the retrospective and heterogeneous character of the data, there are indications that the data collected are representative, as they match existing data in many aspects. For example, a mortality of about 15% is described for patients with TTS-CS [3,7]. In the present data, the mortality was 13.7%. Other parameters such as the frequency of ECG changes or an increase in cardiac markers are also consistent with data collected in larger registries [7,13,70,71].

Concerning the question of which device might be appropriate for treating CS in TTS, the following considerations regarding the pathophysiology of TTS and the mechanisms of the devices can be made and are discussed in the literature underlying this review.

The intra-aortic balloon pump (IABP) works with counterpulsation. This therapy is only able to increase the cardiac index to a limited extent and can worsen or even cause LVOTO [72]. Another argument against IABP is that the rapid heartbeat that is frequently present in TTS-CS may be too fast for IABP to follow [73]. Furthermore, some studies suggest that IABP shows no advantage over optimal drug therapy without mechanical circulatory support [11,17].

Veno-arterial extracorporeal membrane oxygenation (v-a ECLS) is often used with the argument that the patient is suffering from respiratory failure on top of uncompensated acute heart failure despite inotropic and ventilatory support. Other reasons for the use of ECLS are that it also can be applied in children or patients who are too short for Impella [74,75].

One reason against using v-a ECLS is that, due to providing retrograde circulatory support, it increases LV afterload, which can further increase the already elevated left ventricular end-diastolic pressure (LVEDP).

The microaxial Impella pump provides physiological cardiac support which improves coronary and end-organ perfusion while simultaneously unloading the left ventricle. The main advantages of the Impella in TTS-CS are that cardiac output can be increased simultaneously with venting of the overloaded LV, thereby improving pulmonary congestion by reducing LV preload and increasing cardiac output regardless of the heart rate. Also, in several reports, the availability of the Impella was given as the reason for choosing this device [26,28,52,53,76,77].

The positive effect of LV unloading becomes apparent when comparing patients with CS treated with ECLS alone or with ECMELLA. LV unloading has been associated with a lower 30-day mortality [22].

In order to interpret the outcome of the different device groups in this analysis, the gathered data must first be critically analysed with regard to their quality and comparability. Some parameters were only reported for a fraction of the patients in the primary literature. Other parameters, such as the categorisation of the trigger as emotional or physical, are not based on hard criteria, but on what the authors of the case reports write. The achievement of complete recovery is also a descriptive parameter, which was collected from the texts of the underlying literature and is not based on defined clinical parameters. However, some of the parameters were well reported: baseline LVEF was available for 76.6% of the patients, and the duration of mechanical circulatory support was available for 91.1% of patients. Other parameters that would have led to a more precise comparability of the groups were, unfortunately, not available at all.

It can be assumed that the patients who received the Impella for the treatment of CS in the context of TTS were equally affected by pump failure, represented by reduced LVEF, which was comparable between the groups; only the IABP group was significantly different with a better LVEF. Therefore, for the other groups, we can assume that the level of cardiac compromise was comparably severe. Additional other parameters such as NTproBNP, elevated troponin and cardiac arrest also indicate a comparable disease severity between the groups. The Impella group showed a significantly shorter duration of mechanical circulatory support. This finding, along with a comparable disease severity within the Impella group, may suggest that left ventricular venting is relevant for a rapid recovery from TTS.

Several publications about TTS-CS demonstrate high survival and recovery rates, also since TTS is, by definition, a form of temporary heart failure [4,7]. Therefore, when dealing with cardiogenic shock in the context of TTS, MCS can be used as part of a bridge-to-recovery concept. As the literature also clearly shows that the complications of MCS increase with time on support, MCS therapy should be administered for as short a time as possible to avoid complications [19]. For this reason, microaxial LVADs may be the best option for treating CS in TTS. It could therefore be assumed that in the present data, with significantly shorter support durations in the Impella group, significantly lower complication rates would also be expected. However, this was not the case. One reason for this could be that the Impella group also included data from the multicentre study of Impella use in TTS-CS by Napp et al., which also provides detailed data on the type and frequency of complications, unlike the majority of the case reports, which rarely focus on or even mention MCS complications. This demonstrates the urgent need for prospective multicentre studies investigating the use of MCS devices in TTS-CS with a critical eye on complications associated with MCS.

The available data are also insufficient to make a statement on the benefit of one device with regard to the recovery presented in the follow-up. A complicating factor here is that the underlying case reports often do not provide any information about the follow-up period or the clinical parameters that demonstrate recovery. In some cases, it was only described that complete recovery was achieved and that the patient was discharged home.

In conclusion, the data suggest that unloading the ballooned ventricle accelerates myocardial recovery, and that preference should be given to devices that increase neither the gradient across the LVOT nor the LVEDP. To provide evidence for the treatment of CS induced by TTS, prospective and multicentre studies investigating different MCS devices as well as mechanical versus conservative shock therapies in this specific setting are urgently needed.

## 5. Limitations

Three main points can be highlighted which limit the results of this systematic review. First, the heterogeneous data base: our systematic review represents the largest population of patients with mechanical circulatory support for CS caused by TTS to date. It is based predominantly on case reports, which are often very detailed but usually focus on a specific question and therefore do not always represent all parameters in precisely the same manner. There are also no standardized criteria for some parameters, such as a patient’s full recovery. Parameters such as the frequency of catheterization prior to MCS implantation clearly show the weakness of an analysis based on such heterogenic baseline data. In the case of an exclusion diagnosis such as TTS, a very high proportion of catheterizations would be expected to rule out ACS. However, catheterization prior to MCS is reported in only 66.9% of cases, which clearly appears too low and exemplifies the limitations of the present work.

Second, this analysis is retrospective and covers a period of 20 years. During this period, MCS technology improved and also became more widely available. This may also have changed the potential patient population in whom MCS could be applied in the setting of TTS-CS. Since the SCAI classification was established in 2019, the SCAI stage was not documented in the individual case reports, not least because the literature included in the present analysis covers a period of 20 years. Furthermore, the retrospective nature of this study poses a challenge for the inclusion of studies. The diagnosis of TTS and CS declared by the authors had to be verified on the basis of the data available in the publications (fulfilment of the InterTak diagnostic criteria and haemodynamic as well as clinical criteria of CS). If no additional information was available that would allow the assessment of the InterTak score to determine whether cardiogenic shock caused by takotsubo syndrome was a plausible diagnosis, studies were not included.

Third, when looking at the differences between the various devices, the following must be considered: there is a high cross-over rate between devices in the data set, which reduces the separation between the groups and biases this review towards finding no difference. Furthermore, the conclusion that MCS with the Impella is significantly better than other MCS strategies was based mainly on the parameters of baseline LVEF and duration of MCS. To confirm that the correlations identified here are causal and to clearly show that left ventricular ventilation leads to faster myocardial recovery, prospective multicentre studies are needed that also consider more precise parameters.

## 6. Conclusions

This systematic review and pooled analysis of individual data involved 124 patients with TTS-CS treated with MCS. Cardiac recovery was shorter with the Impella, with no differences in terms of survival and full recovery of left ventricular function. The superiority of microaxial LVADs over the other devices can merely be assumed on the basis of the shorter duration of mechanical support. Prospective randomized studies are needed to provide recommendations on the use of MCS in TTS-CS.

## Figures and Tables

**Figure 1 jcm-13-00473-f001:**
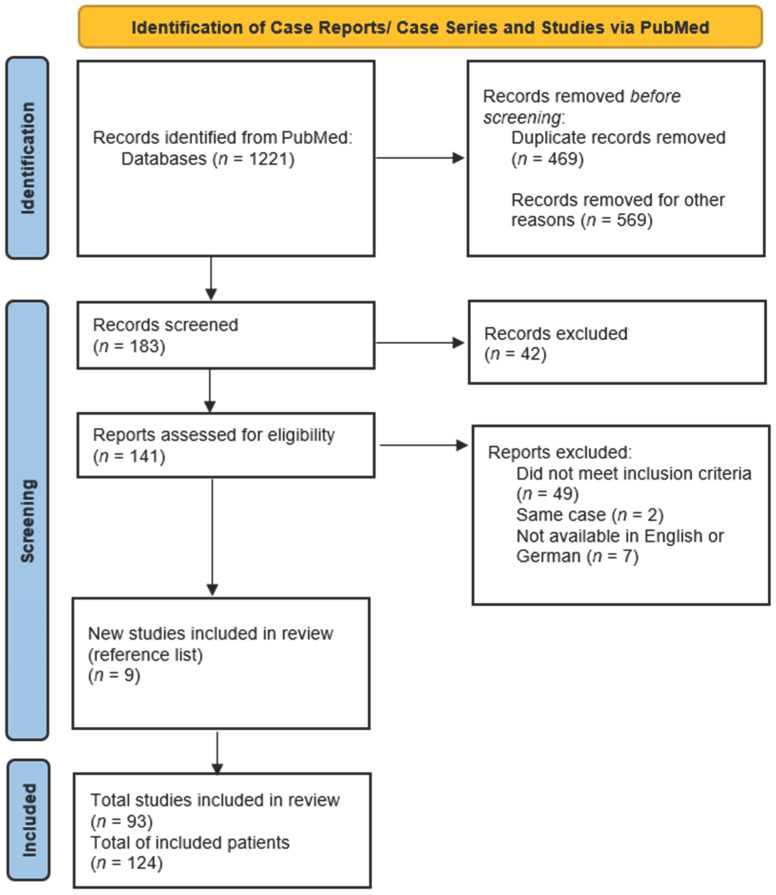
PRISMA (Preferred Reporting Items for Systematic Review and Meta-Analysis) flowchart of the systematic literature review [29].

**Figure 2 jcm-13-00473-f002:**
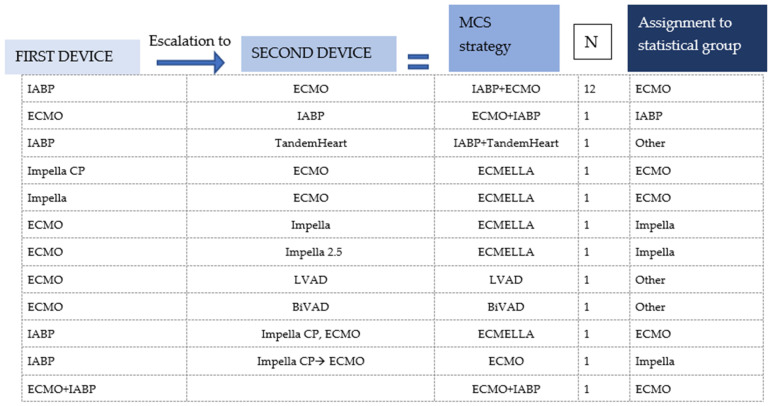
Assignment to statistical group/single-device group of patients who were treated with more than one device and with a combination of two devices or where the first device was removed and replaced by another.

**Figure 3 jcm-13-00473-f003:**
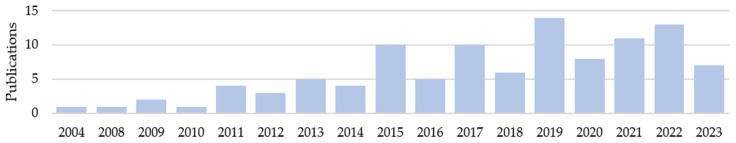
Number of included publications by year.

**Figure 4 jcm-13-00473-f004:**
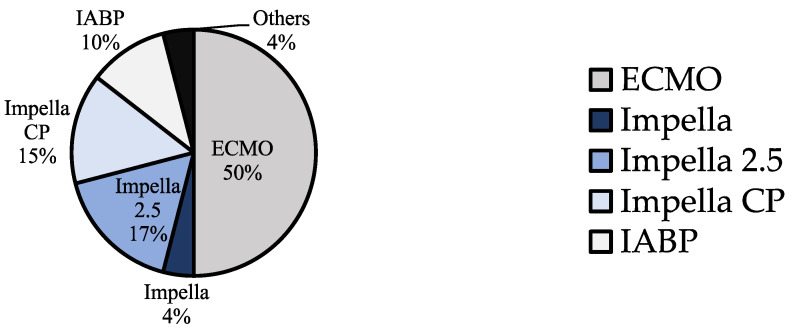
Percentage of the different device groups (ECLS, Impella, Impella 2.5, Impella CP, IABP and Other) in the total collective of 124 patients with TTS and CS treated with MCS.

**Figure 5 jcm-13-00473-f005:**
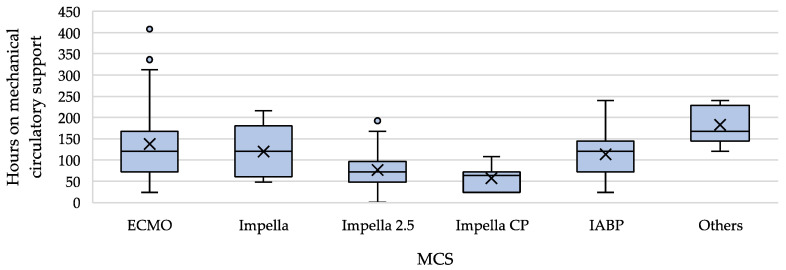
Comparison of mechanical circulatory support duration with median, IQR, outliers and mean (=x). For the number of cases evaluated per group, see Table 3.

**Figure 6 jcm-13-00473-f006:**
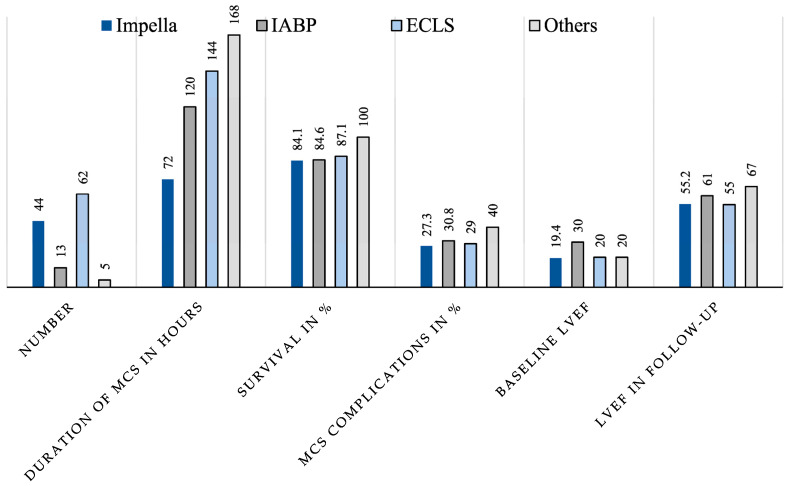
Graphical representation of important parameters for comparing the four different device groups (Impella, IABP, ECMO, Other).

**Table 1 jcm-13-00473-t001:** Demographic characteristics (age and sex). Proportions of the Impella 2.5 and Impella CP subgroups in the Impella group. Percentage of patients in whom inverted TTS was described and percentage of patients in whom the right ventricle was also involved.

	ECMO	Impella	IABP	Others	All	*p*
*N*	62	44	13	5	124	
Impella CP		18 (40.9%)				
Impella 2.5		21 (47.7%)				
Age, median (IQR)	45 (31–56)	68 (55–76)	67.5 (48–69.75)	46 (26–53)	52.2	
Sex category						0.275
Female	42 (67.7%)	34 (77.3%)	12 (92.3%)	4 (80%)	92 (74.2%)	
Male	20 (32.3%)	10 (22.7%)	1 (7.7%)	1 (20%)	32 (25.8%)	
Inverted TTS	17 (27.4)%	4 (9.1%)	2 (15.4%)	1 (20%)	24 (19.4%)	0.099
RV involved	6 (9.7%)	1 (2.3%)	1 (7.7%)	1 (20%)	9 (7.3%)	0.205

**Table 2 jcm-13-00473-t002:** Clinical status before MCS implantation. Metric data are reported as mean with IQR (interquartile range) and number of available data (*n*). Nominal data are given in number of patients and % of the device group.

	ECLS	Impella	IABP	Others	All	*p*
Initial LVEF, median (IQR)	20% (15–24) (*n* = 43)	19.4% (19.4–21.35) (*n* = 38)	30% (22.5–33) (*n* = 9)	20% (9.5–28.5) (*n* = 5)	20% (*n* = 95)	0.015
Inotropes before MCS	47 (75.8%)	35 (79.5%)	11 (84.6%)	5 (100%)	98 (79%)	0.876
Respiratory failure	43 (69.4%)	13 (29.5%)	3 (23.1%)	2 (40%)	61 (49.2%)	<0.001
Arrhythmia	19 (30.6%)	14 (31.8%)	6 (46.2%)	3 (60%)	42 (33.9%)	0.002
ST elevation	24 (38.7%)	22 (50%)	5 (38.5%)	2 (40%)	53 (42.7%)	0.88
ST depression	10 (16.1%)	5 (11.4%)	3 (23.1%)	0	18 (14.5)	0.002
T-wave inversion	16 (25.8%)	6 (13.6%)	4 (30.8%)	0	26 (21%)	0.002
Troponin elevation	44 (71%)	24 (54.5%)	11 (84.6%)	2 (40%)	81 (65.3%)	0.067
Cardiac arrest	23 (37.1%)	9 (20.5%)	3 (32.1%)	2 (40%)	37 (29.8%)	0.081
Angiography before MCS	31 (50%)	41 (93.2%)	10 (76.9%)	1 (20%)	83 (66.9%)	<0.001
NTproBNP, mean	8293.5 (*n* = 13)	13,697.4 (*n* = 5)	1832.0 (*n* = 3)	22,842 (*n* = 1)	9301.85 (*n* = 22)	0.089
BNP, mean	6342.3 (*n* = 8)	946 (*n* = 3)	4900.0 (*n* = 1)	*n* = 0	4873 (*n* = 12)	
BP syst in mmHg, median (IQR) /MAP in mmHg, mean	95 (79–141) (*n* = 35) /59.5 (*n* = 2)	80 (73–100) (*n* = 31) /50 (*n* = 3)	82 (80–115) (*n* = 6) /51.5 (*n* = 4)	74 (*n* = 2) /(*n* = 0)		0.134/0.399
HR median (IQR)	129.5 (100.25–148.75) (*n* = 32)	126 (106.25–136.75) (*n* = 28)	127 (111–147) (*n* = 4)	121 (*n* = 2)		0.933
LVEDP, median (IQR)	25 (*n* = 3)	26 (23–30) (*n* = 11)	12 (*n* = 2)	(*n* = 0)		0.357
Initial lactate, median (IQR)	69.8 (52.6–116.2) (*n* = 18)	44.1 (27.5–57.6) (*n* = 22)	104 (*n* = 2)	(*n* = 0)		0.038
Pheochromocytoma	17	1 (ECMELLA *n* = 2)	(ECLS + IABP: *n* = 1)		19	
Mitral regurgitation	7 (11.3%)	5 (11.4%)	2 (15.4%)	1 (20%)	15 (12.1%)	0.662
Systolic anterior motion	3 (4.8%)	5 (11.4%)	2 (15.4%)	0	10 (8.1%)	
LVOTO	3 (4.8%)	7 (15.9%)	2 (15.4%)	0	12 (9.7%)	0.174
Emotional trigger	16 (25.8%)	17 (38.6%)	15.9 (53.8%)	4 (80%)	44 (35.5%)	0.002
Physical trigger	58 (93.5%)	22 (50%)	11 (84.6%)	4 (80%)	95 (76.6%)	<0.001
Exogenous catecholamine trigger	4 (6.5%)	1 (2.3%)	1 (7.7%)	0	6 (4.8%)	0.583

**Table 3 jcm-13-00473-t003:** Outcome and follow-up data presented as metric data reported as means with IQRs (interquartile ranges) and number of available data (*n*). Nominal data given in number of patients and % of device group. Two side *p*-value of the statistical comparison of the different device groups.

	ECLS	Impella	IABP	Others	All	*p*
Survival in %	54 (87.1%)	37 (84.1%)	11 (84.6%)	5 (100%)	107 (86.3%)	0.855
Hours on MCS, median (IQR)	144 (72–192) (*n* = 55)	72 (48–96) (*n* = 42)	120 (72–144) (*n* = 11)	168 (144–228)	96 (*n* = 113)	<0.001
MCS complications	18 (29%)	12 (27.3%9)	3.8 (30.8%)	2 (40%)	36 (24.2%)	0.927
Follow-up	45 (72.6%)	21 (47.7%)	9 (69.2%)	4 (80%)	79 (63.7%)	0.049
Max. follow-up in weeks, median (IQR)	8 (4–24) (*n* = 45)	4 (2–8) (*n* = 19)	2 (1.75–9) (*n* = 9)	28 (18–44) (*n* = 4)	6 (*n* = 77)	0.01
LVEF in follow-up, median (IQR)	55% (55–65) (*n* = 22)	55.2% (*n* = 22)	61% (*n* = 7)	67% (57–72) (*n* = 5)	58.5% (*n* = 56)	0.262
Full recovery	47 (75.8%)	32 (72.7%)	8 (61.5%)	5 (100%)	92 (74.2%)	0.458

**Table 4 jcm-13-00473-t004:** Frequency of reported complications (multiple mentions per patient possible).

	ECLS	Impella	IABP	Others
Complications under MCS	18 (29%)	12 (27.3%)	4 (30.8%)	2 (40%)
Stroke	1 (1.6%)	0	0	0
Prolonged weaning of MCS	2 (3.2%)	0	0	0
Pericardial tamponade	2 (3.2%)	0	0	0
MR development under MCS	0	1 (2.3%)	0	0
Ventricle rupture	0	1 (2.3%)	0	0
Development of atrial fibrillation under MCS	1 (1.6%)	1 (2.3%)	0	0
Haemolysis	0	3 (6.8%)	0	0
Significant HB drop/bleeding	4 (6.5%)	5 (11.4%)	1 (7.7%)	0
Significant thrombocytopenia	3 (4.8%)	3 (6.8%)	0	0
AKI	3 (4.8%)	3 (6.8%)	0	0
Vascular damage in the lower limb	5 (8.1%)	1 (2.3%)	1 (7.7%)	0
Vascular damage in the upper limb	1 (1.6%)	0	0	0
Death while on support	0	1 (2.3%)	2 (15.4%)	0

## Data Availability

All data are available, see Appendix A.

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
