# Peer review of "Mechanical Circulatory Support Strategies in Takotsubo Syndrome with Cardiogenic Shock: A Systematic Review"

_jcm, 2024, doi:10.3390/jcm13020473_

Round 1
Reviewer 1 Report
Comments and Suggestions for Authors
In this systematic review and meta-analysis, von Mackensen et al. included 93 studies involving 124 patients with Takotsubo syndrome treated with any mechanical circulatory support. They found that time to cardiac recovery was shorter with Impella, with no differences in terms of survival and full recovery of left ventricular function.
Thanks to the authors for their valuable work. The paper is well-written and interesting. Although it might be improved. In particular:
• According to the inclusion criteria, the diagnosis of Takotsubo syndrome was done with the InterTak diagnostic criteria: even though this score is based on data of a robust registry (International Takotsubo Registry), it yields only clinical variables and doesn’t encompass imaging and angiography information which are often necessary for TTS diagnosis. This could have biased the results for missing diagnosis.
• As regards the pre-MCS status of the patients, baseline LVEF is considered a key parameter for evaluating the cardiac compromise. Pulmonary artery catheterization is not performed in any of the groups: apart from LVEF, hemodynamic parameters may help to better identify the correct time and the right patient to implant with MCS. The lack of right heart catheterization makes it difficult to draw conclusion as to the comparability of the groups.
• Only 50% patients underwent coronary angiography before MCS implantation in the ECLS: it must be said that Takotsubo still remains a diagnosis of exclusion so it’s reasonable to rule out obstructive coronary artery disease, especially in those critically ill patients assigned to ECMO group.
• There is a high number of patients escalating from a first device (IABP) to a second device (ECMO): such a high rate of cross-over reduces the separation between the groups biasing the review towards to finding no difference.
• More than 50% of the overall patients were in ECMO group: such a percentage of patients on ECM may alter the outcome because they usually have a longer ICU stay, require a longer need for mechanical ventilation and have a higher rate of in-hospital complications (ventilator-associated pneumonia, vascular complications, etc). In fact, patients on ECLS group spent a median of 144 hours with MCS and the highest rate of MCS complications (table 3). Moreover, complication items are not based on common definitions.
• The achievement of complete recovery is not based on defined clinical parameters
• The superiority of microaxial LVAD is just assumed based on the shorter time of mechanical support in comparison to the other devices: the better clinical course of these patients should have been shown with definite clinical, echocardiographic, and hemodynamic parameters. In light of these considerations, I recommend modifying the conclusions to accurately reflect the validity and applicability of the results.
Author Response
- A) Answers and revisions according to the comments from Reviewer #1:
General comment of Reviewer #1.
In this systematic review and meta-analysis, von Mackensen et al. included 93 studies involving 124 patients with Takotsubo syndrome treated with any mechanical circulatory support. They found that time to cardiac recovery was shorter with Impella, with no differences in terms of survival and full recovery of left ventricular function.
Thanks to the authors for their valuable work. The paper is well-written and interesting. Although it might be improved. In particular:
Our answer:
We would like to thank the reviewer for the summarizing comment and reflections on the presented work.
- Specific comment of Reviewer #1.
According to the inclusion criteria, the diagnosis of Takotsubo syndrome was done with the InterTak diagnostic criteria: even though this score is based on data of a robust registry (International Takotsubo Registry), it yields only clinical variables and doesn’t encompass imaging and angiography information which are often necessary for TTS diagnosis. This could have biased the results for missing diagnosis.
Our answer:
We would like to thank the reviewer for this important comment. It is correct that the InterTak score is comprised solely of clinical parameters and was primarily developed to differentiate Takotsubo syndrome from the important differential diagnosis acute coronary syndrome in the emergency room. Our inclusion criterion was that the authors of the underlying work had diagnosed the patient as having Takotsubo syndrome and that the diagnosis was consistent with the InterTak criteria. (TTS diagnosis consistent with the InterTAK Diagnostic Criteria and designated as such by the authors.)
This means that a large proportion of the diagnoses was established using the Mayo Clinic criteria. However, we did not want to use these as inclusion criteria, as they would have excluded Takotsubo syndrome in the context of underlying neurological diseases and pheochromocytoma.
This inclusion criterion resulted in the exclusion of only a handful of papers. In these few cases, the authors were not sure whether Takotsubo syndrome was the cause of cardiogenic shock. If no additional information was available that would allow assessment of the InterTak score to determine whether cardiogenic shock caused by Takotsubo syndrome was a plausible diagnosis, the studies were not included. In the systematic review we see no risk of bias in the results due to the combination of the InterTak score with the inclusion criterion of the Takotsubo Syndrom diagnosis, which the authors of the primary literature have declared reliable. In future prospective studies, however, this question should definitely be considered and, if necessary, inclusion criteria should be developed that combine the Mayo Clinic criteria and the Intertak criteria.
Our corresponding revisions:
Section: Limitations: Line: 423-434
Second, the analysis is retrospective and covers a period of 20 years. During this period, MCS technology has improved and has also become more widely available. This may also have changed the potential patient population in whom MCS could be applied in the setting of TTS-CS. Since the SCAI classification was established in 2019, the SCAI stage was not documented in the individual case reports, not least because the literature included in the present analysis covers a period of 20 years. Furthermore, the retrospective nature of the study poses a challenge for the inclusion of the studies. The diagnosis of TTS and CS declared by the authors had to be verified on the basis of the data available in the publications (fulfilment of the InterTak diagnostic criteria and hemodynamic as well as clinical criteria of CS). If no additional information was available that would allow assessment of the InterTak score to determine whether cardiogenic shock caused by Takotsubo syndrome was a plausible diagnosis, the studies were not included.
- Specific comment of Reviewer #1.
As regards the pre-MCS status of the patients, baseline LVEF is considered a key parameter for evaluating the cardiac compromise. Pulmonary artery catheterization is not performed in any of the groups: apart from LVEF, hemodynamic parameters may help to better identify the correct time and the right patient to implant with MCS. The lack of right heart catheterization makes it difficult to draw conclusion as to the comparability of the groups.
Our answer:
It is correct and unfortunate that the primary data from the systematic literature review is limited in some cases. We considered baseline left ventricular ejection fraction as the most important parameter. It was largely evaluated. Based on the finding that their multivariable logistic regression analysis identified left ventricular ejection fraction as the only significant predictor of cardiogenic shock, Stiermaier et al. (1) conclude that patients with Takotsubo syndrome who are at risk of developing cardiogenic shock may be identified by a severely impaired left ventricular function as their investigations showed in multivariable logistic regression analysis that left ventricular ejection fraction was identified as the only significant predictor of cardiogenic shock.
References:
1: Stiermaier, T., et al., Incidence, determinants and prognostic relevance of cardiogenic shock in patients with Takotsubo cardiomyopathy. Eur Heart J Acute Cardiovasc Care, 2016. 5(6): p. 489-496.
Our corresponding revisions:
Section: Results: Line: 186-194
The quality of the case-related data in the publications varied widely. In particular, some articles were dedicated to specific aspects, e.g. echocardiographic control or therapy of the underlying condition, and therefore provided little general data. Overall, an almost complete data set could be collected for demographics, baseline LVEF and the duration of mechanical support. Especially information regarding the follow-up and the clinical status prior to the establishment of MCS, e.g. blood pressure, heart rate, catecholamines before MCS, varied greatly in terms of availability and quality. Unfortunately, important hemodynamic parameters such as right heart catheter findings or parameters of extended cardiac monitoring were not provided in the underlying literature.
Section: Disscussion: Line: 371- 374
However, some of the parameters were well reported: baseline LVEF, which was available for 76.6% of the patients, and the duration of mechanical circulatory support, which was available for 91.1% of patients. Other parameters that would have led to a more precise comparability of the groups were, unfortunately, not available at all.
- Specific comment of Reviewer #1.
Only 50% patients underwent coronary angiography before MCS implantation in the ECLS: it must be said that Takotsubo still remains a diagnosis of exclusion so it’s reasonable to rule out obstructive coronary artery disease, especially in those critically ill patients assigned to ECMO group.
Our answer:
We fully agree with your comment. The difficulty here was that, in some of the case reports, the diagnosis was established using the Mayo Clinic criteria (which require catheterization), but no further information was given as regards the catheterization or the exclusion of acute coronary syndrome. In order not to confound the results, we decided to include the performance of cardiac catheterization in the pooled analysis only if it was explicitly mentioned in the underlying literature. This once again demonstrates the difficulty of working with a large amount of very heterogeneous basic literature.
Our corresponding revisions:
Section: Limitations: Line: 414-425
Three main points can be highlighted which limit the results of the systematic review. First, the heterogeneous data base: Our systematic review represents the largest population of patients with mechanical circulatory support in cardiogenic shock caused by Takotsubo syndrome to date. It is based predominantly on case reports, which are often very detailed but usually focus on a specific question and therefore do not always represent all parameters in precisely the same manner. There are also no standardized criteria for some parameters, such as a patient’s full recovery. Parameters such as the frequency of catheterization prior to MCS implantation clearly show the weakness of an analysis based on such heterogenic baseline data. In the case of an exclusion diagnosis such as TTS, a very high proportion of catheterizations would be expected to rule out ACS. However, catheterization prior to MCS is reported in only 66.9%, which clearly appears too low and exemplifies the limitations of the present work.
- Specific comment of Reviewer #1.
There is a high number of patients escalating from a first device (IABP) to a second device (ECMO): such a high rate of cross-over reduces the separation between the groups biasing the review towards to finding no difference.
Our answer:
We appreciate this valuable addition. A corresponding comment will be added to the limitations.
Our corresponding revisions:
Section: Limitations: Line 438-445
Third, when looking at the differences between the various devices, the following must be considered: There is a high cross-over rate between devices in the data set, which reduces the separation between the groups and biases the review towards finding no difference. Furthermore, the conclusion that MCS support with Impella is significantly better than other MCS strategies was based mainly on the parameters baseline LVEF and duration of MCS. To confirm that the correlations identified here are causal and to clearly show that left ventricular ventilation leads to faster myocardial recovery, prospective multicentre studies are needed that also consider more precise parameters.
- Specific comment of Reviewer #1.
More than 50% of the overall patients were in ECMO group: such a percentage of patients on ECM may alter the outcome because they usually have a longer ICU stay, require a longer need for mechanical ventilation and have a higher rate of in-hospital complications (ventilator-associated pneumonia, vascular complications, etc). In fact, patients on ECLS group spent a median of 144 hours with MCS and the highest rate of MCS complications (table 3). Moreover, complication items are not based on common definitions.
Our answer:
Thank you for pointing this out. Without question, there is certainly a distortion due to the unequal size of the device groups. The aim of the review is to summarize the existing literature on this topic; therefore, all articles that met the inclusion criteria were included, even if this resulted in very unevenly sized groups. Another point is that the underlying case reports are dedicated primarily to very different questions. For example, some case reports discuss exceptionally complicated cases, while others focus on issues such as diagnosis and thus barely address the challenges of circulatory support. Once again, it is evident that there is an urgent need for data to be collected in a standardized manner and, ideally, prospectively.
The items for the assessment of complications are very strictly based on the underlying literature and therefore do not correspond to general criteria. We have added a corresponding note.
Our corresponding revisions:
Section: Outcome and follow-up: Line 281-285
Unfortunately, it was hardly ever reported whether complications occurred during MCS therapy. Furthermore, the recorded complications are not based on standardized definitions, but rather on what the authors of the primary literature report. In total, MCS complications were reported in 36 patients (29%). The occurrence of complications did not differ significantly between the groups (p=0.927).
- Specific comment of Reviewer #1.
The achievement of complete recovery is not based on defined clinical parameters.
Our answer:
We thank you for your valuable comment. Since, in some case reports, the only statement made about the follow-up was that complete recovery was achieved, we decided to record this even though it is merely a descriptive parameter and not a defined clinical parameter. This limitation is already mentioned in the discussion and in the limitations.
Section: Discussion: Line: 368
The achievement of complete recovery is also a descriptive parameter, which was collected from the texts of the underlying literature and is not based on defined clinical parameters.
Section: Limitations: Line: 416
There are also no standardized criteria for some parameters, such as a patient’s full recovery.
- Specific comment of Reviewer #1.
The superiority of microaxial LVAD is just assumed based on the shorter time of mechanical support in comparison to the other devices: the better clinical course of these patients should have been shown with definite clinical, echocardiographic, and hemodynamic parameters. In light of these considerations, I recommend modifying the conclusions to accurately reflect the validity and applicability of the results.
Our answer: We share this understanding of the presented results. We have added a conclusion at the end of the text in order to clearly reflect the validity and applicability of the work presented. This is intended to clearly categorize the work and its results.
Our corresponding revisions:
Section: Conclusion: Line: 448-453
Conclusion:
This systematic review and pooled analysis of individual data involved 124 patients with TTS-CS treated with MCS. Cardiac recovery was shorter with Impella, with no differences in terms of survival and full recovery of left ventricular function. Superiority of microaxial LVADs over the other devices can merely be assumed on the basis of the shorter duration of mechanical support. Prospective randomized studies are needed to provide recommendations on the use of MCS in TTS-CS.
- B) Answers and revisions according to the comments from Reviewer #2:
General comment of Reviewer #2.
In this meta-analyis the authors compared different MCS strategies in cardiogenic shock (CS) patients with TakoTsubo Syndrome (TTS). They described that eventually the IMPELLA device is superior to others with respect to an early recovery of LVEF.
Comments and questions of this reviewer are as follows:
Our answer:
We thank the reviewer for the general and specific comments which we believe have improved our revised paper significantly.
Specific comment of Reviewer #2.
- CS in TTS is associated with relatively low mortality (86% survival), which is in complete contrast to CS of other origin (40-50% despite the use of MCS)
Our answer: In fact, all publications dealing with cardiogenic shock caused by Takotsubo syndrome show significantly lower mortality rates than for cardiogenic shock in the context of other underlying diseases. The mortality rate for cardiogenic shock caused by Takotsubo syndrome is usually reported as 15-20% (1-3), with individual studies reporting rates of up to 23% (4). Overall, however, survival is significantly better than with cardiogenic shock, e.g., in the context of myocardial infarction. Takotsubo syndrome is, by definition, temporary, and spontaneous remission often occurs in less severe forms. In cardiogenic shock caused by Takotsubo syndrome, it is also possible that myocardial function recovers more rapidly than in other forms of cardiogenic shock, so that death due to complications such as end-organ hypoperfusion is less likely to occur. Cardiogenic shock caused by Takotsubo syndrome also occurs in patients without previous illnesses who, unlike patients who suffer from cardiogenic shock in the context of a myocardial infarction, for example, do not suffer from any comorbidities. Patients with cardiogenic shock caused by Takotsubo syndrome are also comparatively younger; in our systematic review, the median age is only 52.2 years. In the literature on cardiogenic shock caused by Takotsubo syndrome available to us, there is no clear explanation for this lower mortality in TTS-CS than in other forms of CS.
References:
1: Schneider, B., et al., Complications in the clinical course of tako-tsubo cardiomyopathy. Int J Cardiol, 2014. 176(1): p. 199-205.
2: Vallabhajosyula, S., et al., Cardiogenic Shock in Takotsubo Cardiomyopathy Versus Acute Myocardial Infarction: An 8-Year National Perspective on Clinical Characteristics, Management, and Outcomes. JACC Heart Fail, 2019. 7(6): p. 469-476.
3: Stiermaier, T., et al., Incidence, determinants and prognostic relevance of cardiogenic shock in patients with Takotsubo cardiomyopathy. Eur Heart J Acute Cardiovasc Care, 2016. 5(6): p. 489-496.
Corresponding section in the manuscript:
Section: Discussion: Line: 320-325
The patient population with Takotsubo syndrome is very heterogeneous, both in terms of concomitant diseases and the triggers of TTS. Some patients develop TTS within the scope of a severe noncardiac condition and therefore have a critical status simply because of their underlying disease. Other patients, however, develop TTS within the scope of an emotional stressor and are physically healthy until they develop TTS.
Specific comment of Reviewer #2.
- How was CS diagnosed?
Our answer: Initially, we included all articles in which the authors had declared cardiogenic shock as confirmed in the context of cardiogenic shock caused by Takotsubo syndrome. While collecting data from the primary literature, the diagnosis of cardiogenic shock was classified as plausible if the following criteria were present:
- Hemodynamic instability in the form of systolic blood pressure below 90 mmHg or need for catecholamines to maintain a systolic pressure above 90 mmHg. Heart rate over 100 bpm or severely reduced left ventricular ejection fraction with strong blood pressure fluctuations (pheochromocytoma).
- Additionally, clear clinical signs of shock such as pulmonary congestion, impaired end-organ perfusion, for example in the form of impaired/altered mental status, oliguria, centralization with cold sweaty extremities, marbling of the skin, increased serum lactate.
In their article "Incidence, determinants and prognostic relevance of cardiogenic shock in patients with Takotsubo cardiomyopathy" (1), Stiermaier et al. conclude that patients at risk of developing cardiogenic shock may be identified by means of a severely impaired left ventricular function. Using multivariable logistic regression analysis, their investigations showed that left ventricular ejection fraction was identified as the only significant predictor of cardiogenic shock by Takotsubo syndrome.
In the cohort we presented, baseline left ventricular ejection fraction was £20%, which is well below that reported in the aforementioned publication (28.8 ± 8%). It can therefore be assumed that the diagnosis of cardiogenic shock in the context of Takotsubo syndrome is plausible.
References:
1: Stiermaier, T., et al., Incidence, determinants and prognostic relevance of cardiogenic shock in patients with Takotsubo cardiomyopathy. Eur Heart J Acute Cardiovasc Care, 2016. 5(6): p. 489-496.
Corresponding section in the manuscript:
Section: Limitations: Line: 432-436
Furthermore, the retrospective nature of the study poses a challenge for the inclusion of the studies. The diagnosis of TTS and CS declared by the authors had to be verified on the basis of the data available in the publications (fulfilment of the InterTak diagnostic criteria and hemodynamic as well as clinical criteria of CS).
Specific comment of Reviewer #2.
3: Are SCAI stages in retrospect available?
Our answer: Thank you for this important question. Unfortunately, it was not possible to retrospectively assess the SCAI shock stages based on the included literature. The classification includes too many parameters that were not regularly documented in the case reports on which this paper is based. Also, the SCAI stage was not documented in the individual case reports. This is certainly due to the fact that the SCAI classification was established in July 2019. In contrast, the literature included in the present analysis was published over the past 20 years.
Our corresponding revisions:
Section: Limitations: Line: 427-438
Second, the analysis is retrospective and covers a period of 20 years. During this period, MCS technology has improved and has also become more widely available. This may also have changed the potential patient population in whom MCS could be applied in the setting of TTS-CS. Since the SCAI classification was established in 2019, the SCAI stage was not documented in the individual case reports, not least because the literature included in the present analysis covers a period of 20 years. Furthermore, the retrospective nature of the study poses a challenge for the inclusion of the studies. The diagnosis of TTS and CS declared by the authors had to be verified on the basis of the data available in the publications (fulfilment of the InterTak diagnostic criteria and hemodynamic as well as clinical criteria of CS). If no additional information was available that would allow assessment of the InterTak score to determine whether cardiogenic shock caused by Takotsubo syndrome was a plausible diagnosis, the studies were not included.
Specific comment of Reviewer #2.
- Is it possible that patients treated with MCS devices were in a low SCAI shock stage (A or B) when they received the device or is it the relatively beneficial course of CS in TTS that explains the low mortality?
Our answer: We thank the reviewer for this important question. The severity of cardiogenic shock varies considerably in Takotsubo syndrome. However, we consider it unlikely that patients with SCAI stage A were included. Stage B patients will, at best, only be sporadic among the included cases. Excluding patients with pheochromocytoma, the median systolic blood pressure was 82mmHg and the median heart rate was 121 bpm. Respiratory failure was found in almost half of the patients at presentation, often with a need for mechanical ventilation. Since cardiac arrest occurred in a total of 29.8% of patients, cardiogenic shock SCAI stage E was, by definition, present in a non-negligible proportion of the cohort. We do not attribute the low mortality to a low SCAI stage. The reported mortality is highly consistent with the mortality recorded in other studies/registries in patients with cardiogenic shock caused by Takotsubo syndrome. We therefore assume that the comparatively low mortality associated with cardiogenic shock in Takotsubo syndrome can be explained by the specific mechanisms and courses of TTS (see answer 1.)
Corresponding section in the manuscript:
Section: Discussion: Line: 332-336
Despite the retrospective and heterogeneous character of the data, there are indications that the data collected are representative, as they match existing data in many aspects. For example, a mortality of about 15% is described for patients with TTS-CS [3, 7]. In the present data, mortality was found to be 13.7%.
Specific comment of Reviewer #2.
- Main draw back for clinical relevance of these data are the heterogeneity of patent cohorts as well as the retrospective and descritpive design. Accordingly, a clear conclusion or even an indirect recommendation for one or the other device is incorrect
Our answer: We completely agree with the reviewer’s comment. We have adjusted any ambiguous statements. In order to clearly and unambiguously categorise the work, its findings and their validity, we have added a conclusion.
Our corresponding revisions:
Section: Conclusion: Line: 448-453
Conclusion:
This systematic review and pooled analysis of individual data involved 124 patients with TTS-CS treated with MCS. Cardiac recovery was shorter with Impella, with no differences in terms of survival and full recovery of left ventricular function. Superiority of microaxial LVADs over the other devices can merely be assumed on the basis of the shorter duration of mechanical support. Prospective randomized studies are needed to provide recommendations on the use of MCS in TTS-CS.
Specific comment of Reviewer #2.
- The use of any of the available devices, which have only recently shown no effect at all in prospective randomized investigations in CS of other origin than TTS, cannot be recommended on a routine basis in CS/TTS patients
Our answer: We agree with the reviewer. Even though the recommendation for the use of mechanical circulatory support in cardiogenic shock was recently changed from class IIb to class IIa in the 2021 Guidelines of the European Society of Cardiology for Acute Heart Failure, the level of evidence still remains a consensus opinion. We therefore agree that no premature recommendations should be made on the use of mechanical circulatory support in cardiogenic shock caused by Takotsubo syndrome. It is again clearly stated in the conclusion that there is no evidence to date to recommend the use of mechanical circulatory support systems in Takotsubo syndrome.
Our corresponding revisions:Section: Conclusion: Line: 448-453
Conclusion:
This systematic review and pooled analysis of individual data involved 124 patients with TTS-CS treated with MCS. Cardiac recovery was shorter with Impella, with no differences in terms of survival and full recovery of left ventricular function. Superiority of microaxial LVADs over the other devices can merely be assumed on the basis of the shorter duration of mechanical support. Prospective randomized studies are needed to provide recommendations on the use of MCS in TTS-CS.
- A prospective randomized use vs. non-use of MCS devices in CS/TTS is the only way to find out
- this should be clearely stated and discussed in this paper
Our answer: In addition to emphasizing the urgent need for multicenter, prospective studies in the discussion, we have now added a conclusion to our paper which once again makes it clear that only randomized, prospective studies can answer the question whether and which mechanical circulatory support devices should be used in cardiogenic shock caused by Takotsubo syndrome.
Our corresponding revisions:
Section: Discussion Line: 410-413
Section: Conclusion: Line: 448-453
Conclusion:
This systematic review and pooled analysis of individual data involved 124 patients with TTS-CS treated with MCS. Cardiac recovery was shorter with Impella, with no differences in terms of survival and full recovery of left ventricular function. Superiority of microaxial LVADs over the other devices can merely be assumed on the basis of the shorter duration of mechanical support. Prospective randomized studies are needed to provide recommendations on the use of MCS in TTS-CS.
- C) Answers and revisions according to the comments from Reviewer #3:
General comment of Reviewer #3:
From the point of view of the Cochrane tools for systematic review and meta-analysis, the article does not follow any of the primary requirements for meta-analysis to be undertaken.
The introduction needs to be shorter, showing an expressive rationale that justifies the methodological path. The methodology needs to be more adequate. The statistics do not include meta-analysis. The methods should be reviewed, or the title should be changed to a narrative or integrative review. The conclusion of the article needs to be included at the end.
Our answer:
We would like to thank the reviewer for this valuable comment.
We agree with the reviewer that our systematic review is not a meta-analysis in the sense of a Cochrane meta-analysis. To carry out such an analysis, the results of individual studies, particularly randomized controlled trials, would have to be available and compared or analyzed together. The literature on the present topic of mechanical circulatory support in cardiogenic shock caused by Takotsubo syndrome consists mainly of case reports and a limited number of smaller case series. As there are currently no clear recommendations on the therapeutic approach to cardiogenic shock caused by Takotsubo syndrome and on the use of mechanical circulatory support in this setting, we considered it appropriate to conduct a systematic literature review. We did this using the Preferred Reporting Items for Systematic and Meta-Analyses (PRISMA). In the methods section, you will find a statement on following the PRISMA guidelines when conducting the literature review.
An overview of the literature on the use of mechanical circulatory support in cardiogenic shock caused by Takotsubo syndrome reveals that a wide variety of support systems are used. However, there are no specific recommendations as to which device should be used when. Given the scarcity of data on the use of different mechanical circulatory support strategies in cardiogenic shock caused by Takotsubo syndrome, we conducted a systematic literature review and then pooled and analyzed the individual data, in particular to identify trends and the need for prospective multicentre randomized studies. This paper is therefore a pooled analysis of individual data (meta-analysis of individual data).
As we agree with the reviewer that this paper does not fulfil the criteria of a meta-analysis in the sense of a Cochrane meta-analysis, we will adjust the title accordingly.
To ensure that the conclusions of this paper do not exceed the limitations of the underlying literature and methods, we have made major revisions to the manuscript.
Corresponding section of the manuscript:
Section: Methods, Line: 104-108
The review was conducted and reported in accordance with the Preferred Reporting Items for Systematic Reviews and Meta-Analyses (PRISMA) guidelines [29]. Prior to the literature review, the concept, inclusion criteria, research question and hypothesis were defined. The aim was to compare the different devices used so far as MSC strategies in the context of TTS-CS in the existing literature.
PRISMA Flow diagram for systematic reviews [1].
References:
Page, M.J., et al., The PRISMA 2020 statement: an updated guideline for reporting systematic reviews. BMJ, 2021. 372: p. n71.
Moher, D., et al., Preferred reporting items for systematic reviews and meta-analyses: the PRISMA statement. PLoS Med, 2009. 6(7): p. e1000097.
Our corresponding revisions:
Section: Titel: Line : 1-3
Mechanical Circulatory Support Strategies in Takotsubo Syndrome with Cardiogenic Shock: A Systematic Review.
Section: Conclusion: Line: 445-450
Conclusion:
This systematic review and pooled analysis of individual data involved 124 patients with TTS-CS treated with MCS. Cardiac recovery was shorter with Impella, with no differences in terms of survival and full recovery of left ventricular function. Superiority of microaxial LVADs over the other devices can merely be assumed on the basis of the shorter duration of mechanical support. Prospective randomized studies are needed to provide recommendations on the use of MCS in TTS-CS.
Section: Limitations, Line: 412-443
Three main points can be highlighted which limit the results of the systematic review. First, the heterogeneous data base: Our systematic review represents the largest population of patients with mechanical circulatory support in cardiogenic shock caused by Takotsubo syndrome to date. It is based predominantly on case reports, which are often very detailed but usually focus on a specific question and therefore do not always represent all parameters in precisely the same manner. There are also no standardized criteria for some parameters, such as a patient’s full recovery. Parameters such as the frequency of catheterization prior to MCS implantation clearly show the weakness of an analysis based on such heterogenic baseline data. In the case of an exclusion diagnosis such as TTS, a very high proportion of catheterizations would be expected to rule out ACS. However, catheterization prior to MCS is reported in only 66.9%, which clearly appears too low and exemplifies the limitations of the present work.
Second, the analysis is retrospective and covers a period of 20 years. During this period, MCS technology has improved and has also become more widely available. This may also have changed the potential patient population in whom MCS could be applied in the setting of TTS-CS. Since the SCAI classification was established in 2019, the SCAI stage was not documented in the individual case reports, not least because the literature included in the present analysis covers a period of 20 years. Furthermore, the retrospective nature of the study poses a challenge for the inclusion of the studies. The diagnosis of TTS and CS declared by the authors had to be verified on the basis of the data available in the publications (fulfilment of the InterTak diagnostic criteria and hemodynamic as well as clinical criteria of CS). If no additional information was available that would allow assessment of the InterTak score to determine whether cardiogenic shock caused by Takotsubo syndrome was a plausible diagnosis, the studies were not included.
Third, when looking at the differences between the various devices, the following must be considered: There is a high cross-over rate between devices in the data set, which reduces the separation between the groups and biases the review towards finding no difference. Furthermore, the conclusion that MCS support with Impella is significantly better than other MCS strategies was based mainly on the parameters baseline LVEF and duration of MCS. To confirm that the correlations identified here are causal and to clearly show that left ventricular ventilation leads to faster myocardial recovery, prospective multicentre studies are needed that also consider more precise parameters.

Reviewer 2 Report
Comments and Suggestions for Authors
In this meta-analyis the authors compared different MCS strategies in cardiogenic shock (CS) patients with TakoTsubo Syndrome (TTS). They described that eventually the IMPELLA device is superior to others with respect to an early recovery of LVEF
Comments and questions of this reviewer as as follows:
1. CS in TTS is associated with relatively low mortality (86% survival), which is in complete contrast to CS of other origin (40-50% despite the use of MCS)
2. How was CS diagnosed?
3: Are SCAI stages in retrospect available?
4. Is it possible that patients treated with MCS devices were in a low SCAI shock stage (A or B) when they received the device or is it the relatively beneficial course of CS in TTS that explains the low mortality?
5. Main draw back for clinical relevance of these data are the heterogeneity of patent cohorts as well as the retrospective and descritpive design. Accordingly, a clear conclusion or even an indirect recommendation for one or the other device is incorrect
6. The use of any of the available devices, which have only recently shown no effect at all in prospective randomized investigations in CS of other origin than TTS, cannot be recommende on a routine basis in CS/TTS patients
7. A prospective randomized use vs. non-use of MCS devices in CS/TTS is the only way to find out
8. this should be clearely stated and discussed in this paper
Reviewer 3 Report
Comments and Suggestions for Authors
From the point of view of the Cochrane tools for systematic review and meta-analysis, the article does not follow any of the primary requirements for meta-analysis to be undertaken.
The introduction needs to be shorter, showing an expressive rationale that justifies the methodological path. The methodology needs to be more adequate. The statistics do not include meta-analysis. The methods should be reviewed, or the title should be changed to a narrative or integrative review. The conclusion of the article needs to be included at the end.
​
Round 2
Reviewer 2 Report
Comments and Suggestions for Authors
The authors have sufficiently answered the comments and criticism of this reviewer
Author Response
We would like to thank Reviwer for his kind feedback on our point-by-point response.
Reviewer 3 Report
Comments and Suggestions for Authors
Most conditions were completed; please remove the meta-analysis description from the text.
Author Response
We would like to thank the reviewer for his comment and feedback on our point-by-point response! All text passages have now been adjusted accordingly.
Our corresponding revision:
Line 1:
Type of Systematic Review and Meta-Analysis.
Section: Abstract, Line 22-24
Methods: A systematic literature research and meta-analysis of individual data was performed in MEDLINE/PubMed according to PRISMA guidelines. Our research considered original works until 31 July 2023.
Section: Abstract, Line 34-35:
Conclusions: Though the Impella treatment is new, our meta-analysis may show a significant benefit of Impella compared to other MCS strategies for cardiogenic shock in Takotsubo syndrome.
Section: Introduction, Line: 101-104:
To recognise the trends and outcomes of this currently rapidly growing population of patients with TTS-CS treated with MCS and to compare the different support systems that can be applied, we performed a systematic review and meta-analysis of the existing literature.
Section: Discussion, Line: 385-387
In order to interpret the outcome of the different device groups in this meta-analysis, the gathered data must first be critically analysed with regard to their quality and comparability